# Probing Inter-modality: Visual Parsing with Self-Attention for Vision-Language Pre-training

Hongwei Xue[1]*, Yupan Huang[2]*, Bei Liu[3], Houwen Peng[3], Jianlong Fu[3], Houqiang Li[1], Jiebo Luo[4]

[1]University of Science and Technology of China, Hefei, China,
[2]Sun Yat-sen University, Guangzhou, China,
[3]Microsoft Research, Beijing, China,
[4]University of Rochester, Rochester, NY
[1]gh051120@mail.ustc.edu.cn, [2]huangyp28@mail2.sysu.edu.cn,
[3]{bei.liu, houwen.peng, jianf}@microsoft.com,
[1]lihq@ustc.edu.cn, [4]jluo@cs.rochester.edu

## Abstract

Vision-Language Pre-training (VLP) aims to learn multi-modal representations from image-text pairs and serves for downstream vision-language tasks in a fine-tuning fashion. The dominant VLP models adopt a CNN-Transformer architecture, which embeds images with a CNN, and then aligns images and texts with a Transformer. Visual relationship between visual contents plays an important role in image understanding and is crucial for inter-modal alignment learning in VLP. However, CNNs have limitations in visual relation learning due to local receptive field's weakness in modeling long-range dependencies. Thus the two objectives of learning visual relation and inter-modal alignment are encapsulated in the same Transformer network. Such design might restrict the inter-modal alignment learning in the Transformer by neglecting the specialized characteristic of each objective. To tackle this challenge, we propose a fully Transformer visual embedding for VLP to better learn visual relation and further promote inter-modal alignment. Specifically, we propose a metric named Inter-Modality Flow (IMF) to measure the interaction between vision and language (i.e., inter-modality). We also design a novel masking optimization mechanism named Masked Feature Regression (MFR) in Transformer to further promote the inter-modality learning. To the best of our knowledge, this is the first study to explore the benefit of Transformer for visual feature learning in VLP. We verify our method on a wide range of vision-language tasks, including Image-Text Retrieval, Visual Question Answering (VQA), Visual Entailment and Visual Reasoning. The result shows that our approach not only outperforms the state-of-the-art VLP models, but also exhibits superiority on the IMF metric.

## 1  Introduction

Vision-Language Pre-training (VLP) has shown great benefits for Visual-Language (VL) tasks such as Visual Question Answering (VQA), Visual Entailment, etc., in many recent works [9, 21, 31, 32, 36, 40, 42, 51]. VLP is designed to learn vision and language (VL) joint representation and alignment with a huge number of image-text pairs. It provides a powerful initialization of multi-modal representation for downstream VL tasks in a fine-tuning way. In this paper, we explore how to improve the multi-modal representation which is the key challenge in VLP from the perspective of intra-modal and inter-modal learning.

---

*This work was performed when Hongwei Xue and Yupan Huang were visiting Microsoft Research as research interns.

35th Conference on Neural Information Processing Systems (NeurIPS 2021).

Existing works use image features as visual tokens and learn their alignments with language tokens using a multi-modal Transformer. Three main types of image representation are commonly used in VLP: region feature, grid feature and patch projection. Most VLP models [9, 42, 40] extract region-based image features with an off-the-shelf object detector [34]. Each visual token in VLP corresponds to a pre-defined region feature. Some recent works directly learn grid features from images by CNNs to lift restrictions of bounding boxes and pre-defined object categories [22, 21]. They train CNN and multi-modal Transformer in an end-to-end fashion to enable visual features optimized for pre-training objectives. In addition to use CNNs for visual feature learning, a recent work ViLT [28] directly input image patch projections into the multi-modal Transformer to achieve the fastest inference speed with the lightest VLP architecture.

In vision-language (VL) tasks, the relation between visual concepts is crucial. For example, given a question "what is the man doing?" for an image of a surfing man, a VL model is expected to infer the relation of "surfing" from object "man" and object "surfboard". However, existing three types of image representations fail to model the intra-vision relation. For both region feature and patch projection, each unit (i.e., bounding box or patch) is independent while global relations are not encoded in the visual embedding. For grid feature learned by CNNs, the local receptive field in the convolution layer results in local features of neighbor regions. Thus the two objectives of VLP, i.e., learning visual relation and inter-modal alignment, are encapsulated in the multi-modal Transformer network. Such design might restrict the inter-modal alignment learning in the Transformer by ignoring the specialized characteristic of each objective. On the other hand, for language, texts are highly structured and relations are explicitly provided by grammar. This inconsistent representation of different modalities distracts the multi-modal Transformer from the inter-modal alignment.

To better learn visual relation and further promote inter-modal alignment, we propose a fully Transformer VLP model which adopts self-attention for visual feature learning. Self-attention breaks the spatial inductive bias and enables long-range global relation learning of visual features. Thus, multi-modal Transformer can be specialized for multi-modal joint learning. In self-attention mechanism, each visual token is an approximate weighted mixture of all tokens. The higher weight indicates higher dependency. We name this way of image feature learning as visual parsing. As visual parsing provides dependencies of each visual token pair, inter-modality learning can be further promoted by masking visual tokens with high dependency, forcing the multi-modal Transformer to have more attention on the language side. To the best of our knowledge, our proposed Masked Feature Regression (MFR) is first ranking-based masking mechanism in VLP that leverages the intrinsic properties of visual extractor. To promote the inter-modality learning, we propose the Inter-Modality Flow (IMF) metric based on Attention Flow [1] to measure the fusion of inter-modality in VLP. IMF aims to quantify the information flow between the two modalities.

To verify the effectiveness of our approach, we conduct experiments on a wide range of vision-language tasks that are highly related to visual relation understanding and inter-modal reasoning. Our approach not only outperforms the state-of-the-art VLP models, but also shows superiority on the proposed IMF metric. We also conduct extensive ablation studies to demonstrate the effectiveness of our proposed self-attention visual parsing and parsing-based masking mechanism. We thoroughly probe the inter-modality learning in VLP from the perspective of information flow and data distribution. Our probing reveals how vision and language fuse with each other.

Our contributions in this paper are summarized as follows:

1 We are the first to adopt self-attention to learn visual features for VLP, aiming to promote inter-modality learning in multi-modal Transformer. Our model outperforms existing works on a wide range of vision-language tasks.

2 We propose a novel Inter-Modality Flow (IMF) metric to measure and reveal vision and language fusion in VLP.

3 We design the first ranking-based masking mechanism for visual self-attention module to further promote inter-modality learning, verified by well-designed ablation studies.

## 2 Related Work

**Multi-Modal Pre-training.** Pioneering works of Vision-Language Pre-training like ViLBERT [36] and LXMERT [42] adopt two-stream architecture, which utilizes two separate Transformers to encode image and text modalities, and a third Transformer for multi-modal fusion. Recent works using

single-stream architecture directly fuse the two modalities in a Transformer to automatically learn inter-and-intra modality fusion [40, 32, 31, 9]. The majority of existing works advocate the single-stream designs, which show stronger performance and include fewer parameters than two-stream models. For above reasons, we mainly focus on one-stream architecture in this paper.

Single-stream architectures typically employ Transformer to learn multi-modal contextualized features based on the singular embedding of each modality. For vision part, most VLP works use Bottom-Up and Top-Down attention [2] to extract region-level visual features by a Faster R-CNN [39] detector pre-trained on Visual Genome dataset [29]. Region-level features may miss the contextual information out of the bounding boxes and often suffer from low quality, noisy, and over-sampled boxes. To overcome the restrictions of region-based image features, SOHO [21] designs an end-to-end VLP pipeline which directly learns image embedding at pixel-level by a CNN. This end-to-end learning paradigm does not require bounding box annotations and it enables inference 10 times faster than region-based approaches. A recent work ViLT [28] directly inputs patch projections into a multi-modal Transformer to achieve a fast inference speed. While ViLT has performance gaps with existing methods on some downstream tasks like VQA.

**Vision Transformer.** As convolutional network only captures information in a local window, it will miss correlations of long-range features. To improve the capability of encoding distant dependencies or heterogeneous interactions, some works complement CNNs by extending self-attention modules [17, 44]. The augmentation by self-attention also benefits general visual feature extraction. This kind of method has been applied for the object detection task [8, 52, 47].

The pioneering work of Vision Transformer (ViT) [13] totally abandons convolution and applies a Transformer architecture on image patch projections for image classification. Inspired by ViT, there are some recent works studying vision Transformer for a broad range of vision tasks such as image classification [35, 43], object detection [4], semantic segmentation [50]. In this paper we adopt [35] as our visual backbone. Since CNN and Transformer favor different kinds of optimizer [48], our fully Transformer structure makes it easier to train and fine-tune. We utilize unified optimizer for vision Transformer and multi-modal Transformer.

## 3 Approach

### 3.1 Self-Attention Visual Parser

Despite grid features overcome the restrictions of region-based image features and keep all visual information in images [21], CNN-based image feature learning remains challenging when utilized to vision-language pre-training. Specifically, convolution network tends to focus on local regions in images, adding the learning of global visual relation to the multi-modal Transformer. Instead of concentrating on inter-modal interaction learning, multi-modal Transformer has to learn intra-model interactions in visual modality as well. For the language side, texts are highly structured and they explicitly provide some relations from grammar. For example, prepositions and verbs often indicate the relations of objects. Thus, the two modalities are fed into the multi-modal Transformer at different information levels. This inconsistency makes Transformer tend to learn stand-alone representation of each modality. As a result, the multi-modal Transformer is distracted from focusing on learning inter-modal alignment.

Inspired by Vision Transformer (ViT) [13], we apply self-attention for visual embedding in vision-language pre-training. To learn visual features, we adopt a vision Transformer $VT$ with parameters $\theta$:

$$\mathcal{V} = VT(I, \theta) \in \mathbf{R}^{m \times c}, \tag{1}$$

where $I$ is the input image. $m$ denotes the number of visual features, and $c$ is the dimension of hidden states. To reduce computational complexity, we adopt Swin Transformer [35], which has a hierarchical structure. Although shifted windows makes the receptive field smaller than the whole image, the large window size and gradual down-sampling mechanism make the spatial inductive bias much lower than CNN. We also explicitly get the correlations of visual tokens. In the last layer of vision Transformer, the correlation of each token pair can be modeled by attention weights which track how features get mixed.

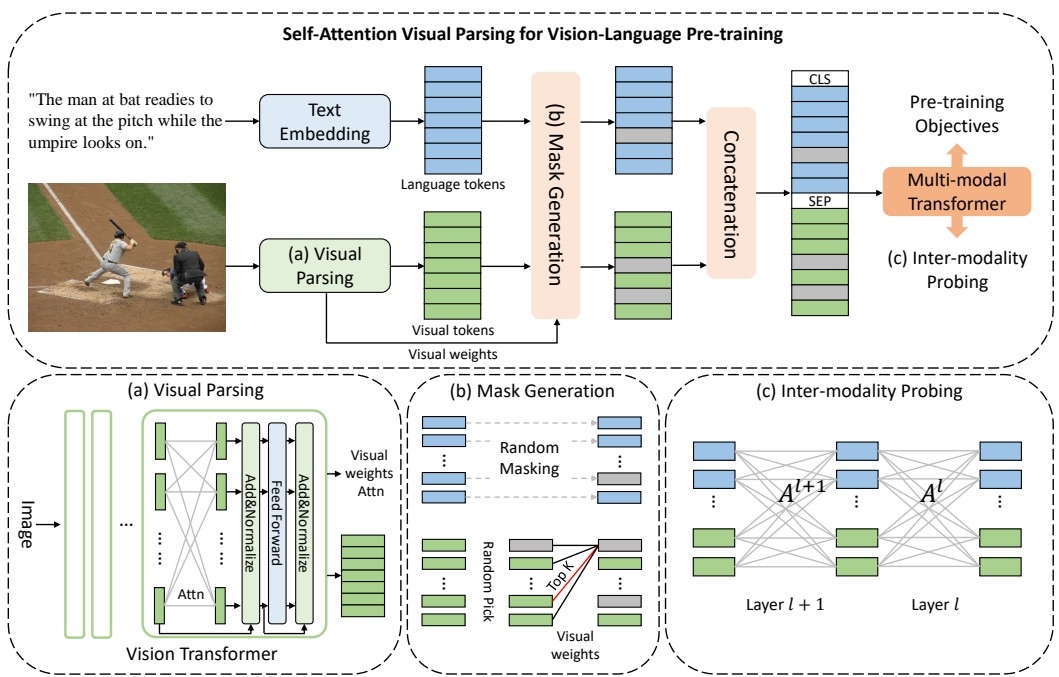

Figure 1: Pipeline of our Self-Attention Visual Parsing for Vision-Language Pre-training. (a) The Visual Parsing module applies a vision Transformer to learn visual representations. (b) The Mask Generation module generates masks for visual and language tokens, which are concatenated and passed to a Multi-modal Transformer for joint embedding learning. (c) Inter-modality Probing measures the vision-language alignment with our proposed IMF metric.

## 3.2 Inter-modality Probing

In single-stream structure for Vision-Language Pre-training, a multi-modal Transformer is used to fuse the vision and language modality. To quantify the interactions between two modalities, we propose a metric named Inter-Modality Flow (IMF) to probe how vision and language fuse together.

Transformer is a stack of self-attention modules and feed forward neural networks. The interactions of image-and-text are learnt in self-attention modules. Following Attention Flow [1], we first compute values in layer $l$ after excluding shared operations on each tokens as:

$$T_{l+1} = T_l + W_{att}^l T_l, \tag{2}$$

where $T_l$ and $T_{l+1}$ are the input and output of layer $l$. $W_{att}^l$ is layer $l$'s attention weight. To account for residual connections, weights of mixture can be represented as:

$$A^l = 0.5(I + W_{att}^l). \tag{3}$$

To track down the information propagating in Transformer layers, we first calculate attention flows of arbitrary two layers, e.g., layer $i$ and layer $j$ where $j \geq i$ :

$$A^{i,j} = \prod_{a=j}^{i} A^a. \tag{4}$$

This calculation sums over all possible paths between two layers, and $A^{i,j}$ is the attention weight matrix of layer $i$'s input to layer $j$'s output. Specifically, when $j = i$, $A^{i,i}$ represents attention weights in layer $i$. To measure the interactions between two modalities, we propose an Inter-Modality Flow $F_{inter}^{i,j}$ and it is computed as the proportion of inter-modal attention among all attentions:

$$F_{inter}^{i,j} = A_{inter}^{i,j}/(A_{inter}^{i,j} + A_{intra}^{i,j}), \tag{5}$$

where $A_{inter}^{i,j}$ is the summation over all attentions of visual and language inter-modality features, while $A_{intra}^{i,j}$ sums over one modality, vision or language:

$$A_{inter}^{i,j} = \sum_{\phi_{inter}} A_{x,y}^{i,j}; \quad A_{intra}^{i,j} = \sum_{\phi_{intra}} A_{x,y}^{i,j}, \tag{6}$$

where $\phi_{inter} = \{x \in V, y \in L \ or \ x \in L, y \in V\}$ and $\phi_{intra} = \{x, y \in V \ or \ x, y \in L\}$. $L$ and $V$ are token index set of vision and text. We calculate $F_{inter}^{i,j}$ on pre-training data to probe how the two modalities interact with each other.

## 3.3 Pre-training Pipeline

**Model Overview.** Figure 1 shows the overview of our vision-language pre-training framework. Our model is composed of a vision Transformer and a multi-modal Transformer. The vision Transformer takes an image as input and outputs the visual tokens $\mathcal{V} = \{v_1, v_2, ..., v_i, ...v_m\}$. To encode spatial information of images, we utilize 2-D position embedding computed by sine function following other works [8, 37, 13]. We apply a Linear layer and Layer Normalization [3] to embed the vision tokens. For the input sentence, we follow BERT [11] to tokenize and get word embeddings $\mathcal{W}$. We concatenate vision and language tokens ($\mathcal{V}$ and $\mathcal{W}$) to form an input sequence for multi-modal learning. Similar to other VLP models, we add two special tokens [CLS] and [SEP] into the input sequence to indicate classification position and the separation of two modalities, respectively. A multi-layer Transformer is adopted to take the joint vision-language input, and outputs the attended features. In order to distinguish it from the vision Transformer $VT$, we call it multi-modal Transformer $MT$.

We adopt three pre-training tasks in our model: Masked Language Modeling (MLM), Image-Text Matching (ITM) and Masked Feature Regression (MFR). Among them, MLM and ITM are two commonly used pre-training task and MFR is a novel task which is proposed to mask visual tokens with similar or correlated semantics in our framework.

**Masked Language Modeling.** We adopt Masked Language Modeling (MLM) following most vision-language pre-training works [9, 21, 28] to predict the ground-truth labels of masked text tokens from the contextualized tokens:

$$\mathcal{L}_{\text{MLM}} = -\mathbb{E}_{(\mathcal{W}, \mathcal{V})} \log p\left(w_i \mid \mathcal{W}_{\backslash i}, \mathcal{V}\right). \tag{7}$$

We adopt the same masking strategy as in BERT, and use a Linear layer as the MLM head to output logits over vocabulary, which are then computed as the negative log likelihood loss for the masked tokens.

**Image-Text Matching.** To learn the inter-modal alignment, we adopt Image-Text Matching (ITM) task for pre-training as in most vision-language pre-training works [9, 21, 28]. With the probability of 0.5, we randomly replace the aligned image to a different image. We use a single linear layer as ITM head to predict logits $y$ over binary class (match or not), and we compute negative log likelihood loss as our ITM loss:

$$\mathcal{L}_{\text{ITM}} = -\mathbb{E}_{(\mathcal{W}, \mathcal{V})} \log p(y \mid \mathcal{W}, \mathcal{V}). \tag{8}$$

We also design a vision-language token alignment (VLA) task inspired by the word region alignment objective in [9, 28]. Our VLA loss optimize the Optimal Transport (OT) distance approximated by IPOT [46]. Following [9, 28], we add the VLA loss multiplied by 0.1 to the ITM loss.

**Masked Feature Regression.** Without bounding boxes, random masking for visual feature regression loses its effectiveness as the model will directly copy from neighbor features [21]. By visual parsing introduced in Section 3.1, correlations of each visual token are explicitly modeled by attention weights. Similar to $A^l$ in Equation 3, we use the last layer's attention weight for masking. We assume that visual tokens of high attentions share similar semantics or correlations. We first randomly pick one visual token to mask, and then mask tokens with top-$k$ attention weights. We apply L2 regression between masked tokens and regressed features:

$$\mathcal{L}_{\text{MFR}} = \sum_{i=1}^{k} \left\| \mathbf{v}_{\mathbf{m}}^{(i)} - r\left(\mathbf{v}_{\mathbf{m}}^{(i)}\right) \right\|_2^2. \tag{9}$$

The vision Transformer and multi-modal Transformer are trained end-to-end with above objectives:

$$\mathcal{L}_{\text{pre}} = \mathcal{L}_{\text{MLM}} + \lambda_1 \mathcal{L}_{\text{ITM}} + \lambda_2 \mathcal{L}_{\text{MFR}} \tag{10}$$

Table 1: Evaluation of image-to-text retrieval (TR) and text-to-image retrieval (IR) on Flickr30K dataset. "-" indicates the detail is not reported.

| | Method | VSE++[14] | SCAN[30] | ViLBERT[36] | Unicoder[31] | UNITER[9] | ViLT[28] | SOHO[21] | Ours |
|---|---|---|---|---|---|---|---|---|---|
| | R@1 | 52.9 | 67.4 | - | 86.2 | 85.9 | 83.7 | 86.5 | **87.0** |
| TR | R@5 | 80.5 | 90.3 | - | 96.3 | 97.1 | 97.2 | 98.1 | **98.4** |
| | R@10 | 87.2 | 95.8 | - | 99.0 | 98.8 | 98.1 | 99.3 | **99.5** |
| | R@1 | 39.6 | 48.6 | 58.2 | 71.5 | 72.5 | 62.2 | 72.5 | **73.5** |
| IR | R@5 | 70.1 | 77.7 | 84.9 | 90.9 | 92.4 | 87.6 | 92.7 | **93.1** |
| | R@10 | 79.5 | 85.2 | 91.5 | 94.9 | 96.1 | 93.2 | 96.1 | **96.4** |

## 4 Experiments

### 4.1 Pre-training Details

We follow the dataset settings of SOHO [21] for pre-training. We focus on in-domain datasets: MSCOCO Captions (MSCOCO) [33] and Visual Genome Dense Captions (VG) [29], which are typical in-domain datasets for many VL downstream tasks. When comparing with UNITER [9], we use its in-domain pre-training results for fairness.

We follow BERT to adopt the WordPiece tokenizer [49] to split a sentence into word tokens. We adopt Swin Transformer for vision Transformer and a 12-layer Transformer for multi-modal Transformer. If not specified, we use $384 \times 384$ as input resolution. This resolution is much lower than the resolution of $600 \times 1000$ or $800 \times 1333$ adopted by most previous works [36, 9, 21, 28]. Increasing the resolution will likely further improve our performance as per the findings by [25]. Our models are initialized based on ImageNet [10] and BERT [11]. We use AdamW optimizers for vision Transformer with learning rate 5e-6 and multi-modal Transformer with learning rate 5e-5 empirically. Empirically, we set the k in Equation 9 as 7. We set the $\lambda_1$ and $\lambda_2$ in Equation 10 as 1 and 0.01 respectively. Our model is pre-trained on 8 NVIDIA Tesla V100 GPUs with a batch size of 2048. Following SOHO [21], the learning rate is warmed up for the first 500 iterations. The training process takes 40 epochs until convergence and the learning rate decays by 10 times at 25th and 35th epoch. To reduce memory cost, we pair an image with four texts in each batch, including two matched pairs and two unmatched pairs. MLM and MFR tasks are only calculated with the matched image-text pairs.

### 4.2 Downstream Tasks

We evaluate our method by fine-tuning the pre-trained model on downstream vision-language tasks. As we particularly focuses on visual relation and inter-modal learning, we choose four tasks related to visual relation understanding and inter-modal reasoning: Image-Text Retrieval, Visual Question Answering (VQA), natural language for visual reasoning (NLVR), and fine-grained visual reasoning (Visual Entailment). We compare our model with several task-specific and pre-training models. Among them, SOHO [21] and ViLT [28] are end-to-end VLP models like ours. SOHO, ViLT and our model adopt CNN, Linear projection and Transformer for visual embedding learning, respectively.

**Image-Text Retrieval.** Image-text retrieval aims to retrieve the most relevant text from candidate images, or vice versa. Image-text retrieval includes two sub-tasks: image-to-text retrieval (TR) and text-to-image retrieval (IR). We follow the same practice as SOHO to conduct image-text retrieval for fair comparisons. During training, we construct image-text pairs in a mini-batch by sampling aligned pairs from ground-truth annotations, and unmatched pairs from other captions within the mini-batch. To predict whether an image-text pair is aligned or not, we use the joint embedding representation of the [CLS] token from Transformers to perform binary classification. Since the binary classification objective of image-text retrieval model is consistent with the image-text matching (ITM) task in pre-training stage, we initialize the task-specific head from the pre-trained ITM head for better initialization. We adopt AdamW optimizer with a learning rate of 5e-5. The mini-batch size is set to 32. We train 10 epochs until convergence and decay the learning rate by half at 5th epoch empirically.

Experiment results on Flickr30k [38] are shown in Table 1. Our model outperforms ViLT and SOHO under all metrics on Flickr30k. The promising results of our model on image-text retrieval indicate the advantage of our fully Transformer architecture for learning cross-modal alignment.

Table 2: Evaluation of VQA on VQA 2.0 dataset. "-" indicates the detail is not reported.

| Model | test-dev | test-std |
|---|---|---|
| MUTAN[5] | 60.17 | - |
| BUTD[2] | 65.32 | 65.67 |
| Unified VLP [51] | 70.50 | 70.70 |
| ViLBERT[36] | 70.55 | 70.92 |
| VisualBERT[32] | 70.80 | 71.00 |
| VLBERT[40] | 71.79 | 72.22 |
| LXMERT[42] | 72.42 | 72.54 |
| UNITER[9] | 72.70 | 72.91 |
| ViLT-B/32-Aug[28] | 70.94 | |
| SOHO[21] | 73.25 | 73.47 |
| Ours | **74.00** | **74.17** |

Table 3: Evaluation of Visual Reasoning on NLVR$^2$ dataset.

| Model | dev | test-P |
|---|---|---|
| MAC[23] | 50.80 | 51.40 |
| Text Only[41] | 50.90 | 51.10 |
| Image Only[41] | 51.60 | 51.90 |
| CNN+RNN[41] | 53.50 | 52.40 |
| MaxEnt[41] | 54.10 | 54.80 |
| VisualBERT[32] | 67.40 | 67.00 |
| LXMERT[42] | 74.90 | 74.50 |
| UNITER[9] | 75.85 | 75.80 |
| ViLT-B/32-Aug[28] | 75.24 | 76.21 |
| SOHO[21] | 76.37 | 77.32 |
| Ours | **77.61** | **78.05** |

Table 4: Evaluation of Visual Entailment on SNLI-VE. "-" indicates the detail is not reported.

| Method | EVE-Image[45] | e-SNLI-VE-2.0[12] | UNITER[42] | SOHO[21] | Ours |
|---|---|---|---|---|---|
| val | 71.56 | 73.02 | 78.59 | **85.00** | 84.75 |
| test | 71.16 | - | 78.28 | 84.95 | **85.08** |

**Visual Question Answering.** Visual Question Answering (VQA) aims to give an answer for a question and an image. VQA task is typically formulated as a classification problem according to the majority of existing works. We adopt a multi-layer perception head to output logits from the [CLS] token. Following [27], we optimize the model by a binary cross-entropy loss on 3,192-way classifications. We fine-tune for 100 epochs with a batch size of 512 until convergence. We keep the optimizer the same as in the pre-training stage, and we decay the learning rate by 10 at the 65th and 75th epochs empirically.

We present the experimental results on VQA v2.0 dataset in Table 2. Our model obtains 0.75% and 0.7% absolute gains on test-dev and test-std splits over SOHO respectively. The results validate the effectiveness of modeling the intra-modality visual information with self-attention mechanism to facilitate intelligent visual question answering.

**Visual Reasoning.** Visual Reasoning with Natural Language (NLVR) aims to predict whether a text is related to a given pair of images. With the requirement of comparing two image-text pairs, NLVR further focuses on the compositional visual reasoning abilities of relations, comparisons, and quantities. For this task, we evaluate our model on NLVR$^2$ dataset [41]. As there are two input images instead of one input image used in the pre-training setup, we follow the *pair* method introduced in UNITER [9] to input two image-text pairs to Transformer and get two embedding vectors from [CLS] tokens. Then we learn a classifier that takes the concatenation of the embedding vectors to infer "true" or "false" by a cross-entropy loss. We fine-tune for 80 epochs with a batch size of 512 until convergence. We decay the learning rate by 2 at the 60, 65, 70, 75, 78, 79th epochs empirically.

Results are shown in Table 3. We observe a clear improvement of our model over previous state-of-the-art model SOHO, where 1.24% and 0.73% absolute accuracy improvements are obtained on the val and test splits respectively. Over 2% improvements are made over ViLT. Our promising results demonstrate our advantage in compositional visual reasoning by promoting inter-modal learning with a fully Transformer architecture.

**Visual Entailment.** Visual Entailment (VE) is a fine-grained visual reasoning task to infer whether an image semantically entails a text. To classify the more fine-grained relationship than NLVR between an image and a text pair, VE aims to infer the image-to-text relationship to be true (entailment), false (contradiction) or neutral. For this task, we evaluate our model on SNLI-VE dataset [45] which is constructed based on Stanford Natural Language Inference (SNLI) [6] and Flickr30K [38] datasets. We follow [9, 21] to perform the VE task as a three-way classification problem. The model predicts the scores of each class by a Linear layer on the representation output by the Transformer from the [CLS] token. We fine-tune the model for 25 epochs with a batch size of 512 until convergence. The learning rate is initialized as 5e-5, and decayed by 10 at the 15th, 20th and 23th epochs empirically.

Table 5: Ablation Study on the effectiveness of self-attention visual parsing and the proposed Masked Feature Regression (MFR) objective. A 3-layer Transformer is adopted for multi-modal Transformer. For the backbone of visual embedding, R50, R101, Swin-T and $T_{Lan}$ indicate ResNet 50, ResNet 101, Swin-tiny Transformer and language-specific Transformer respectively. FLOPs and Parameter numbers of different visual embedding backbones are listed.

| Backbone | FLOPs (G) | Params (M) | Objectives | $F_{inter}^{1,3}$ | VQA dev | VQA std |
|---|---|---|---|---|---|---|
| R50 | 4.1 | 25.6 | MLM+ITM | 0.437 | 64.65 | 65.12 |
| R50 | 4.1 | 25.6 | MLM+ITM+MFR$_{CNN}$ | 0.439 | 64.89 | 65.46 |
| R101 | 7.8 | 44.5 | MLM+ITM | 0.441 | 65.44 | 65.71 |
| Swin-T | 4.5 | 28.0 | MLM+ITM | 0.461 | 67.13 | 67.40 |
| Swin-T | 4.5 | 28.0 | MLM+ITM+MFR$_{Rand}$ | 0.463 | 67.24 | 67.41 |
| Swin-T | 4.5 | 28.0 | MLM+ITM+MFR | 0.474 | 67.74 | 67.90 |
| Swin-T+T$_{Lan}$ | 4.7 | 48.0 | MLM+ITM+MFR | 0.485 | 67.96 | 68.23 |

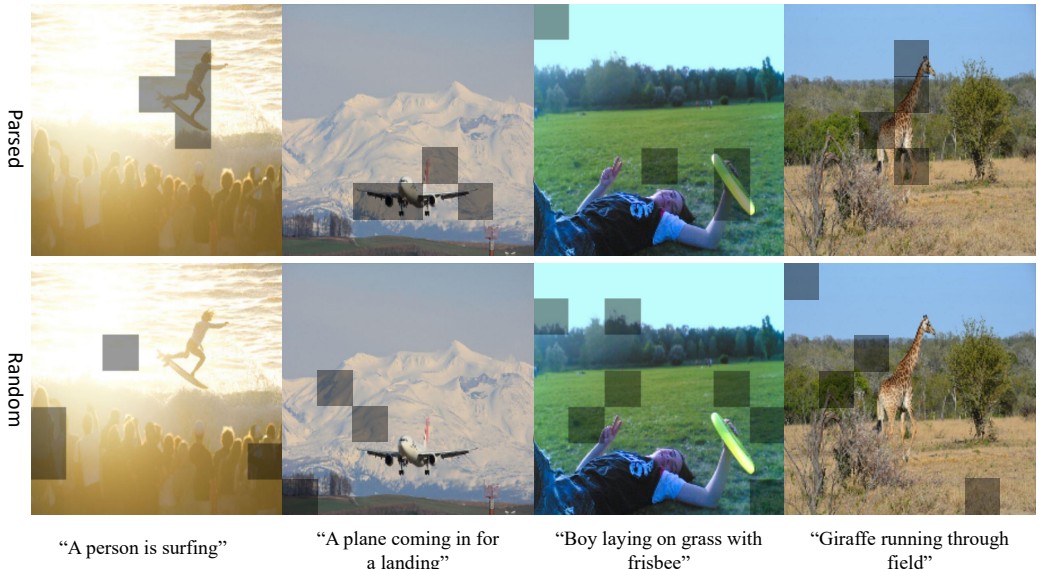

"A person is surfing"    "A plane coming in for a landing"    "Boy laying on grass with frisbee"    "Giraffe running through field"

Figure 2: Visualization of our visual parsing. The first row is masking based on visual attention weights. while the second row is random masking. We randomly sample and show a related caption of each image.

Table 4 compares models on SNLI-VE dataset. Here we see that our model and SOHO are significantly better than UNITER with about 6% absolute gains on accuracy due to our end-to-end training architecture. Our method is comparable with SOHO on the val split and is slightly better than SOHO on the test split.

## 4.3 Ablation Study

To validate the effectiveness of our visual parsing for vision-language pre-training, we conduct ablation studies on the VQA task. To compare CNN and Transformer on visual embedding, we adopt ResNet 50 and 101 [19] which respectively have comparable and much larger parameters and FLOPs compared with Swin-Tiny Transformer. For multi-modal Transformer, we use a 3-layer Transformer. We use the same pre-training and VQA settings as in Section 4.1, except for image resolution. We set resolution as $224 \times 224$ in all ablation studies for fair comparison. We also study the proposed Masked Feature Regression (MFR) objective to verify the effectiveness of masking based on the parsing attention weights. Specifically, we design four settings: without MFR, random masking (MFR$_{Rand}$), masking based on parsing weights (MFR), and a variant (MFR$_{CNN}$) designed for CNN which measures correlations by cosine distance. We set $k$ in Equation 9 as visual tokens number $m$ times the probability of random masking $p$: $k = m * p$.

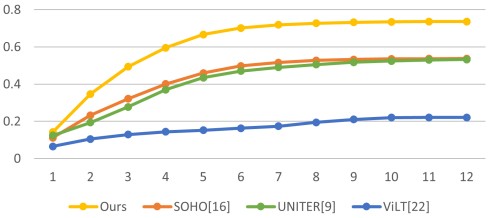
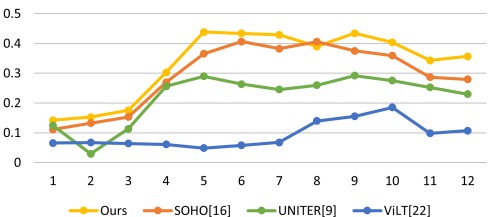

Figure 3: Inter-Modality Flow (IMF) from the input to each layer of the multi-modal Transformer $F_{inter}^{1,j}, j \in [1, 12]$ calculated on ViLT [28], SOHO [21], UNITER [9] and our model.

Figure 4: Inter-Modality Flow (IMF) within each layer of the multi-modal Transformer $F_{inter}^{i,i}, i \in [1, 12]$ calculated on ViLT [28], SOHO [21], UNITER [9] and our model.

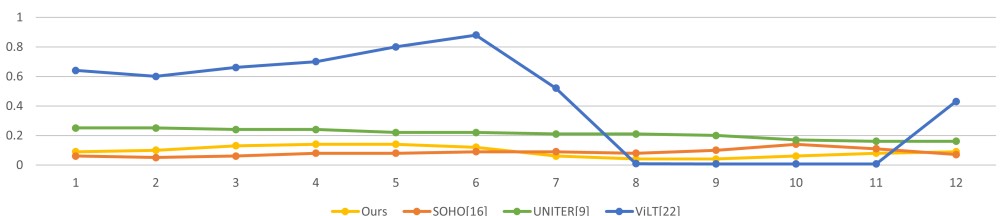

Figure 5: NMI scores calculated on Visual Genome for multi-modal fusion probing from the view of data distribution. A smaller NMI value indicates a higher fusion degree. We compare NMI scores of ViLT [28], SOHO [21], UNITER [9] and our model.

Results are presented in Table 5. The Inter-Modality Flows of all Swin-Tiny backbones are higher than ResNet 50 and ResNet 101. This shows that vision Transformer provides better visual embedding for inter-modality learning compared with CNNs. Although the parameters and FLOPs of the ResNet 101 backbone is more than 1.5 times of our Swin-Tiny Transformer backbone, the latter achieves better result by about 2% on the VQA score. Inter-Modality Flow verifies the superiority of self-attention visual embedding in promoting inter-modality learning. From this view, we further try to add a language-specific Transformer $T_{Lan}$ which process the input language before feeding into cross-modal Transformer. From the results, $T_{Lan}$ can also promote performances. For fair comparison. we do not include $T_{Lan}$ in our final model.

To study the proposed Masked Feature Regression (MFR), the last two lines show that the masking based on the attention weights further promotes the interaction of two modalities. $MFR_{CNN}$ also shows improvement yet less significant than MFR. To intuitively demonstrate the attention-based masking, we sample some results and visualize the mask of visual tokens. Compared with random masking, attention-based masking tends to mask regions with similar semantics. In Figure 2, random masking lacks the ability of preventing the multi-modal Transformer from directly copying from neighbor regions. While attention-based masking mechanism forces the model to refer to the language side for inferring masked features.

### 4.4 Inter-modality Probing

In this section, we aim to verify our model's effectiveness of promoting inter-modality learning and better study how the two modalities fuse together. We choose UNITER [9], SOHO [21], ViLT [28] and our model as models that use region features, grid features, patch projections, features learned by self-attention, respectively. These four models all adopt a 12-layer Transformer for fusion. The fusion of two modalities can be demonstrated from two perspectives: the inter-modal interaction and feature distance of two modalities in the joint space. Thus we do the inter-modality probing by our proposed Inter-Modality Flow (IMF) and distance of data distribution measurement. We will discuss the difference between these two perspectives in this section.

**Inter-Modality Flow.**     Inter-Modality Flow is proposed to quantify interaction between vision and language modalities. Majority of existing works on VLP fuse the vision and language modality

by a multi-modal Transformer. Our proposed Inter-Modality Flow measures the information flow between two modalities from layer $i$'s input to layer $j$'s output. A larger information flow implies more interactions between image and text. We design two Inter-Modality Flow probing tasks layer by layer in the multi-modal Transformer. We calculate the $F_{inter}^{1,j}, j \in [1, 12]$ to track the process of inter-modality learning and $F_{inter}^{i,i}, i \in [1, 12]$ to probe each layer's impact on modality fusion.

From the result shown in Figure 3, our model has larger IMF from the input tokens to every layers consistently. This indicates that there are more interactions between image and text in our model. SOHO and UNITER have comparable IMF. ViLT has much smaller IMF than other methods, as patch projections learn little about visual relationships. The result shown in Figure 4 indicates that the interactions gradually increase in deeper layers. It is worth noting that the interactions decrease a little in the last few layers as each modality optimizes by different pre-training tasks such as MLM and MFR. For the models in these two probing tasks, the IMF is positively correlated with the performance of downstream tasks, which testify to the importance of inter-modality learning in VLP.

**Data Distribution.** Following [7], we study the multi-modal fusion degree of a model from the view of data distribution. In every layer, we extract all visual and language features and apply the k-means algorithm (with k = 2) on features to partition them into two clusters. We measure the difference between the k-means clusters and ground-truth visual/textual clusters via Normalized Mutual Information (NMI) which is an unsupervised metric for evaluating differences between clusters. A larger NMI value implies more significant distinction between two clusters, indicating a lower fusion degree [24]. We average all NMI scores on data samples from Visual Genome.

From Figure 5, SOHO and our model has lower NMI scores than UNITER in every layer consistently. One explanation is that end-to-end training of visual backbone reduces the distinction between two modalities' data distributions. The NMI scores of UNITER gradually decrease while SOHO and our model's NMI score fluctuate within small values. In ViLT, layers with large and small NMI both exist. NMI of ViLT is overall larger than the other three models as it lacks visual backbone to optimize visual features towards alignment to word embeddings. As NMI only evaluates differences between data distributions, it lacks the capacity of measuring inter-modal interactions. Thus, our proposed Inter-Modality Flow is better correlated to inter-modal alignment.

## 5   Conclusion

In this paper, we explore the benefit of self-attention for visual feature learning in Vision-Language Pre-training (VLP). We explore CNN's limitation in intra-vision modality learning by introducing fully Transformer for visual embedding in VLP. We also design a novel masking optimization mechanism named Masked Feature Regression (MFR) in Transformer to further promote the inter-modality learning. To verify MFR and probe the mechanism of inter-modal alignment, we propose Inter-Modality Flow (IMF) to measure the interaction between vision and language modalities. In the future, we will further explore the different properties of various modalities and design a unified framework for inter-modality interaction [20]. Another potential direction is to involve more than two modalities in inter-modality learning to enable more powerful understanding and interactions, such as image tags [16, 15].

## Acknowledgments and Disclosure of Funding

Funding in direct support of this work: NSF of China under Grant 61836011 and 62021001, GPUs provided by Microsoft Research Asia. Additional revenues related to this work: Internship at Microsoft Research Asia.

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

## A  Experiment Statistics

We summarizes the statistics of all our pre-training and downstream tasks in Table 7.

Table 6: Statistics of different tasks. Notation "*" denotes Karpathy split [26]. Notation "-" denotes not applicable.

| Task | Dataset | Train Split | Test Split | Metric |
|---|---|---|---|---|
| Pre-training | VG [29] | train | - | - |
| | MSOCO [33] | train+restval* | - | - |
| Image-Text Retrieval | Flickr30K [38] | train | test* | Recall@1,5,10 |
| Visual Question Answering | VQA2.0 [18] | train+val | test-dev/test-std | VQA-score [18] |
| Visual Reasoning | NLVR$^2$ [41] | train | dev/test-P | Top-1 Accuracy |
| Visual Entailment | SNLI-VE [45] | train | val/test | Top-1 Accuracy |

## B  Model Size

As performances are strongly related to image size and Transformer token number which affect FLOPs, we consider FLOPs as model size. we list FLOPs of models in detail:

Table 7: FLOPs of Vision-Language pre-training models.

| Visual Feature Type | Model | FLOPs (G) |
|---|---|---|
| Region | ViLBERT | 958.1 |
| | VisualBERT | 425.0 |
| | LXMERT | 952.0 |
| | UNITER | 949.9 |
| | UnicoderVL | 419.7 |
| CNN | SOHO | 125.2 |
| Linear | ViLT | 55.9 |
| Self Attention | Ours | 84.28 |

From the results of above table, we can see that our model has much fewer FLOPs than SOHO and region-based models.

