# OpenReview forum: "Probing Inter-modality: Visual Parsing with Self-Attention for Vision-and-Language Pre-training"
_NeurIPS.cc/2021/Conference — NeurIPS 2021 Poster_

### Official Review · Reviewer_1TrJ · 2021-07-16

**Rating:** 6
**Confidence:** 5

**Summary:**

This paper proposes a transformer based visual backbone for vision and language understanding. It uses a vision transformer backbone (Swin Transformer) to extract visual representation, which is then passed to a mask generation module. The visual tokens(including masked tokens) and text tokens embeddings are fed to a multimodal transformer model, where different pretraining objectives are applied. This architecture achieves state-of-the-art results in VQA, SNLI-VE, NLVR^2, Image-Text Retrieval. Authors propose an Inter Modality Flow metric that measure interaction between the visual and textual modalities and correlates well with the downstream results.

**Limitations And Societal Impact:**

Please consider adding the Broader Impact Statement to the main paper rather than in the Appendix.

**Main Review:**

STRENGTHS

- It is impressive that the model achieves state-of-the-art results with a smaller input image dimension : 384x384 compared to previous works that use much larger image input dimensions like 600×1000 or 800×1333 which were difficult to train end to end due to huge computational cost.
- Results on downstream tasks shows that with limited pretraining data but better visual representations, good accuracy can be achieved.
- The Inter Modality Flow metric seems to correlate well with the downstream results. This can act as a strong validation metric to monitor during pretraining on multimodal datasets.

WEAKNESSES and REMARKS

- It is not clear to me from the ablation studies where the performance improvement is coming from. From Table 4. ablations, switching the backbone to SWIN-T resulted in the biggest improvement in accuracy. Is it because Swin Transformer has already learnt better visual features to start with and hence provides better results when multimodal pretraining is performed with visual features extracted from that? In general, in the vision and language community it has been observed that improved visual backbone helps to achieve superior downstream results on tasks like VQA etc. For example, the recent work [1], shows that a very powerful image backbone can generate representations of a richer collection of visual objects and concepts and hence get impressive downstream results in V&L tasks.

      [1] Zhang, Pengchuan, et al. "Vinvl: Revisiting visual representations in vision-language models." Proceedings of the IEEE/CVF Conference on Computer Vision and Pattern Recognition. 2021.

- For fair comparison with other end to end models like SOHO, authors should provide comparative results with SOHO model where it’s CNN encoder is replaced with a Swin Transformer of similar capacity as used in this work.
- It is not clear which Swin Transformer(Swin-T, Swin-S, Swin-B, Swin-L) is used for different experiments reported in Table 1, Table 2, Table 3. It will be better if parameter counts are also reported for different approaches in Table 1, Table 2, Table 3 for fair comparison and understanding whether the improvements correlate to param count.
- MFR(Masked Feature Regression) metric in different forms has already been explored in previous works like VilBERT, UNITER etc. The only difference in this work as I understand is that the visual features are masked by masking all top-k attention weights for a particular randomly picked token. Given that, I am not sure this is a novel contribution.
- The improvement from adding the MFR task is relatively very low (+0.5) improvement over no MFR. Is 0.5 a statistically significant improvement? It is usually better to take average of multiple runs when the gap is too small.
- IMF metric is used for evaluating different models. It is not clear how IMF is used during the training process or how it helps to improve the model : whether it is used during pretraining as a validation metric to pick the best checkpoint or just used was a framework to evaluate different models after they are pretrained? If it is the former, then for fair comparison, IMF metric should also be used during pretraining of UNITER, SOHO etc to understand the value of the metric to give a better pertained model. However, IMF metric is not shown in ablation studies in Table 4, which makes me assume it was not used as a metric to monitor the model during pretraining.
- I am unsure about the added novelty of this work, it seems that majority of the performance improvements come from a better visual encoder backbone, and although the IMF metric shows the model to be better than SOHO/UNITER, it seems much of it is due to using a better vision encoder (Swin Transformer vs R/X-101).


I would be interested to see the discussion and authors' replies to the concerns during the response stage. I am overall leaning towards improving the rating.

**Time Spent Reviewing:**

3

---

> ### Author Response · Authors · 2021-08-10
> **Response to Reviewer 1TrJ**
>
> 1. **Performance gain.** The performance improvement mainly comes from two aspects: a strong visual backbone, and our proposed MFR which uses the attention weight of Transformer to group patches for masking. We agree that visual backbone is very important for VLP. In this work, we explore to analyze the effect of visual backbone. We observe that the performances have strong correlation with multi-modal interaction, which is influenced by visual backbone. VinVL mentioned by the reviewer mainly focus on the vision-only aspect while our work analyzes the multi-modal interaction.
> To better utilize the Transformer module in VLP, we propose MFR, which is the first masking method leveraging the intrinsic properties of the visual extractor. The result also shows the effectiveness of MFR.
>
> 2. **Comparison with e2e models.** As our model structure is similar to Pixel-Bert and SOHO except visual backbone, we make fair comparison on CNN and Transformer in our ablation study. The setting of Swin-T as backbone without MFR is very close to the reviewer's suggestion.
>
> 3. **Model size.** As performances are strongly related to image size and Transformer token number which affect FLOPs, we consider FLOPs as model capacity. we will list FLOPs in detail:
> |Visual Type  |   Model        |      FLOPs (G)|
> |-------------|----------------|----------------|
> | Region             |  ViLBERT       |      958.1 |
> | Region             |  VisualBERT    |      425.0|
> |Region        |  LXMERT        |      952.0|
> | Region            |   UNITER        |      949.9|
> | Region            |   UnicoderVL    |      419.7|
> |CNN            | SOHO     |           125.2|
> |Linear         | ViLT     |           55.9|
> |Transformer    | Ours     |           84.28|
>
>     In Table 1, Table 2, Table 3 we adopt Swin-base. From the results of above table, we can see that our model has much fewer FLOPs than SOHO.
>
> 4. **Novelty of MFR.**
> Our MFR is the first ranking-based masking method leveraging the intrinsic properties of the visual extractor, instead of random masking.
> The pre-training objective of MFR (Masked Feature Regression) is effective in previous region-based pre-training works (e.g., ViLBERT, UNITER). However, MFR is still under exploration in end-to-end grid-based pre-training works. For example, the objective of MPP (Masked Patch Prediction) proposed in ViLT harms downstream tasks' performance [22]. Our ablation experiments also show that masking visual features randomly is useless. To solve this, we propose ranking-based MFR to explore using the intrinsic properties of the visual extractor to mask. Specifically, we leverage the attention scores in vision Transformer. This effective masking method makes the cross-modal Transformer attend more into the language side, leading to better inter-modal learning.
>
> 5. **Significance of performance gain.**
> According to VQA performance improvements of recent works, the number of 0.5 is significant for VQA task. We further conduct ablation study on NLVR2 task and the results are shown below:
> |Method|nlvr dev|nlvr test-p|
> |--------|--------|----------|
> |w/o MFR| 68.12  | 67.52|
> |MFR    | 68.95  |68.50  |
>     We observe much numerical significant improvement of NLVR2 than VQA. Consistent improvements achieved by both tasks validate the effectiveness of our proposed MFR task.
>
>
> 6. **Usage of IMF.**
> The reviewer's assumption is right. IMF is used to measure the influence of different structures on multi-modal interaction. We don't use it during the training process to affect the pre-training model. Thus our comparisons of different pre-trained models are fair.
>
>
> 7. **Novelty.**
> To the best of our knowledge, this is the first work studying the effect of different visual model structures (e.g., bottom-up features, CNN features, image patches, visual tokens) on multi-modal learning.
> Under this motivation, we propose (a) visual parsing and (b) MFR to achieve state-of-the-art results with even smaller image size. We propose (c) IMF as a quantifiable measurement to measure the influence of different structures on multi-modal interaction.
> We believe our work will inspire future discussion and analysis on structure design for vision-language pre-training.
>
>     (a) Visual parsing. This work first explore the vision Transformer for vision-language pre-training. The whole architecture is more consistent with the multi-modal transformers, which simplify the training procedure by unifying optimizers.
>
>     (b) Masked Feature Regression (MFR). Our MFR is the first ranking-based masking method leveraging the intrinsic properties of the visual extractor, instead of random masking. The pre-training objective of MFR (Masked Feature Regression) is effective in previous region-based pre-training works (e.g., ViLBERT, UNITER). However, MFR is still under exploration in end-to-end grid-based pre-training works. For example, the objective of MPP (Masked Patch Prediction) proposed in ViLT harms downstream tasks' performance [22]. Our ablation experiments also show that masking visual features randomly is useless. To solve this, we propose ranking-based MFR to explore using the intrinsic properties of the visual extractor to mask. Specifically, we leverage the attention scores in vision Transformer. This effective masking method makes the cross-modal Transformer attend more into the language side, leading to better inter-modal learning.
>
>     (c) Inter-Modality Flow (IMF). The IMF is a novel metric to reflect the cross-modal interactions of different vision-language frameworks. Our IMF shows high correlation with downstream performances, which validates the importance of inter-modal interaction during pre-training.
>
>
> Last, we will move the Broader Impact Statement and retrieval results to the main paper.

---

### Official Review · Reviewer_9tzA · 2021-07-16

**Rating:** 7
**Confidence:** 4

**Summary:**

This paper provides a novel way of applying masked feature regression (MFR) to transformer-based visual encoders. In addition to this new way of MRM, the paper also provides a quantifiable measurement of how two modalities (image and text) interchange their information with the aids from the attention flow method. This quantity, the inter-modality flow (IMF), is then used to analyze the characteristics of different vision-and-language pretraining models.

**Limitations And Societal Impact:**

I think the authors fairly well addressed the limitations and potential negative societal impact of their work.

**Main Review:**

This paper provides a novel way of applying masked feature regression (MFR) to transformer-based visual encoders. In addition to this new way of MRM, the paper also provides a quantifiable measurement of how two modalities (image and text) interchange their information with the aids from the attention flow method. This quantity, the inter-modality flow (IMF), is then used to analyze the characteristics of different vision-and-language pretraining models.

Question 1. I wonder if MFR can be applied to the features of non-transformer visual encoders (CNNs) by adopting correlations between spatial features other than attention weights. If so, I am eager to see how it can boost the performance of Pixel-BERT, which does not have a visual-related masked modeling objective.

Comment 1. The main model uses a Swin transformer as the visual encoder. Since Swin uses local window attention to build hierarchical structures out of vision transformer, it also has a local receptive field like CNN. Thus, the CNNs' limitations in visual relation learning mentioned in the abstract can also be applied to Swin. I believe this delineation should be revised.

Comment 2. The authors model each layer's inter-modality flow by *indexing* the *contextualized vector sequence* with the indices assigned to each modality before the first attention layer. However, the information of each index is mixed all together by processed through layers after layers. Thus, the vectors can not be considered a hundred percent text modality or a hundred percent image modality. The near-zero NMI at layers 8 to 11 of ViLT (figure 5) makes this speculation more concrete since the near-zero NMI means that two modalities are heavily mingled.

Question 2. To further analyze Comment 2, I am curious about how the indices are assigned to each cluster changes over layer in figure 5. Will the text and image indices be assigned to the same cluster for every layer?

### Remarks
[+] (Clarity) Overall, the paper is well written and easy to understand.
[+] (Originality) I find originality out of both (1) using the attention weight to group the patches to mask and (2) employing the attention flow to analyze the inner interactions of the modalities.
[+] (Significance) The proposed MFR is significant because it liberates vision-and-language pretraining methods' visual masked modeling from its pre-trained object detectors, which are used to obtain the target distributions of masked regions. However, MFR is only applicable to transformer-based visual encoders, limiting its usage for CNN visual encoders. (Question 1)
[-] (Quality) The limitation of CNNs is too exaggerated. (Comment 1)
[-] (Quality) The inter-modality flow should be further analyzed. (Comment 2 and Question 2)

**Time Spent Reviewing:**

6 hours

---

> ### Author Response · Authors · 2021-08-10
> **Response to Reviewer 9tzA**
>
> 1. **Generalization of MFR.**
> The ranking idea in MFR can be applied to the features of CNNs. To validate this, we add an ablation study to adopt correlations between CNNs' spatial features as per your instructive suggestion. More specifically, we use cosine distance to measure correlations and mask top-K visual features like MFR. The results are as follows:
> |Method        |          IMF     |  VQA dev  |   VQA std  |
> |--------------|----------------|-----------|---------|
> |R50           |         0.437   | 64.65    |   65.12 |
> |R50 w/ MFR for CNN  |   0.439   | 64.89    |  65.46   |
>
>     The results show that our ranking-based masking also improves the performances of CNN-based backbone. Thus ranking idea in MFR is a general method and has potential for boosting the performance of PixelBERT.
>     While the improvement is less significant than MFR on Transformer-based visual encoders. The reason might be the fact that spatial correlation is not as direct as attention weights. However, our proposed ranking idea in masking mechanism might encourage future works on designing more effective masking strategy.
>
>
> 2. **Limitation of CNN.**
> We agree with the reviewer's comment on the limitation of CNN and Swin-Transformer. We will revise this delineation in final version.
>
> 3. **Analysis of inter-modality flow.**
>
>     (1) Comment 2: The output vectors can still represent the input tokens of corresponding positions by position embedding and skip connections. This can be illustrated by the usage and effectiveness of MLM and MFR (or other masking modeling objectives) in many VLP models (e.g., ViLBERT, UNITER, SOHO). For example, MLM requires the model to predict masked input tokens by applying regression loss function to output vectors of corresponding position. This assumption is also used in [7].
>
>     (2) Question 2: we assume the reviewer means the ground-truth clusters assignment. The answer is yes. We follow the calculation in [7] and assign the same cluster to text and image indices in each layer.

---

### Official Review · Reviewer_mRRK · 2021-07-16

**Rating:** 6
**Confidence:** 4

**Summary:**

This paper introduces Swin Transformer in the visual domain for vision-language pretraining. Previous VL pretraining models feed image tokens and text tokens together into the cross-modal transformer, and the visual relationship learning and inter-modal alignment are encapsulated in the same transformer network. The authors propose that visual relationship between different visual objects or concepts is important for inter-modal alignment learning, and there should be independent processing of visual relationship in the framework. Motivated by this hypothesis, the authors add a visual-only transformer to learn visual relationship before feeding the image and text tokens together to the cross-modal transformer. In addition, the authors propose a metric named inter-modality flow (IMF) to measure the cross-modal interactions. They also propose masked feature regression as a new pretraining task. Pretraining is conducted on in-domain datasets COCO and Visual Genome. Evaluation is conducted on three downstream tasks VQA, visual reasoning, and visual entailment.

**Ethical Concerns:**

There are no ethical concerns.

**Limitations And Societal Impact:**

This paper does not discuss much on limitations and societal impact. I encourage the authors to add some discussions on the potential limitations, problems, or failure cases of the proposed method.

**Main Review:**

Strengths:

1. The motivation is clear and the method is clearly written.

2. The inter-modality flow is a novel metric to reflect the cross-modal interactions of different vision-language frameworks. The authors have deep discussion into this metric, and compare the IMF of the proposed approach and previous appraoches. The data distribution analysis in page 9 is also interesting and indicates that the proposed model learns better aligned vision-language features.

3. The vision-only structure for learning visual relationship of different objects has not been explored before. This paper makes the first attempt on introducing transformer-based visual relationship reasoning before the cross-modal transformer. I think it might inspire future work, discussions, and analysis.

Weakness:

1. The effectiveness of the swin transformer is not fully validated in the ablation study. To make fair comparisons, at least two experiments should be conducted. (1) Compare swin transformer with other vision transformers, such as ViT, to validate the superior of the swin transformer structure. (2) Compare with a baseline model that does not have the visual parsing module, i.e., the image patches are directly projected into visual tokens and then feed into the cross-modal transformer. This experiment can validate how well the visual parsing module works.

2. Why there is a vision-only transformer, but not a language-only transformer, before the cross-modal transformer. I would expect an ablation study experiment on adding a language parsing module which uses a transformer to process the input sentences. I am curious whether adding the language transformer will help or harm the performance.

3. The pretrained model is not evaluated on the downstream tasks of image captioning and text-image retrieval, which I think are two important downstream tasks for vision-langauge pretraining.

4. In Lines 298-299, "ViLT has much smaller IMF than other methods, as patch projections learn little about visual relationships." I do not totally understand it. I understand that smaller IMF indicates less inter-modality interactions, but how does it related to the visual relationships?


**Time Spent Reviewing:**

3 hours

---

> ### Author Response · Authors · 2021-08-10
> **Response to Reviewer mRRK**
>
> 1. **Ablation on visual backbone.**
>
>     (1) We made some attempts to compare with ViT, but we observes a large performance degradation compared with other models. A recent work [44] also report this problem. We assume that the ViT structure fails to learn visual features suitable for VL tasks. We will study this problem in detail as a future work.
>
>     (2) We would like to clarify that ViLT [22] is such a baseline model that directly projects image patches into visual tokens and feed the tokens into a cross-modal Transformer. We have demonstrated the effectiveness of our visual parsing module in Table 1 and Table 2 by comparing our model with ViLT [22].
>
> ```
> [44] Shen, Sheng, et al. "How Much Can CLIP Benefit Vision-and-Language Tasks?." arXiv preprint arXiv:2107.06383 (2021).
> ```
>
> 2. **Ablation on adding a language Transformer.**
> We did not adopt a language-only Transformer for fair comparisons with previous VL pre-training works (e.g., VisualBERT, VL-BERT, Unicoder-VL, UNITER, SOHO, ViLT), which simply encode language with word embedding.
> According to your suggestion, we conduct an ablation study on adding a language encoding module to process the input sequences with a three-layer Transformer. The results are as follows:
> |Method  |                 IMF   |  VQA dev   |  VQA std  |
> |--------|-----------------------|------------|-------|
> |Ours     |               0.474  | 67.74      | 67.90  |
> |Ours w/ Lan-Transformer |0.485   |67.96       |68.23    |
>
>     We observe VQA and IMF score improvements by adding a language-only Transformer that promotes the inter-modal learning, similar to the function of our proposed visual parsing module. The positive results indicate our model can achieve even higher performance with an extra cost of 20M parameters.
>
> 3. **About downstream tasks.** Thanks for your suggestions. (1) We have evaluated our model on the image-text retrieval task, where our model achieves superior performance over other works. The results are shown in our supplementary material due to limited space, and we will move it to the main paper to emphasize it in the final version. (2) We did not perform image captioning task since we focus on VL understanding tasks instead of VL generation tasks. Besides, our comparing works (i.e., ViLT [22], UNITER [9], SOHO [16]) did not report results on the task of image captioning, so there is no direct comparison with previous works on this task.
>
> 4. **Delineation about ViLT.** In ViLT, as the patch projects learn little about visual relationships, the cross-modal Transformer has to encapsulate the two objectives of learning visual relation and inter-modal alignment in one single module. This will lead the lack of sufficient inter-modality interaction (Line 47-48). The smaller IMF in ViLT proves the above assumption. We will revise our analysis in this part to make it clearer.
>
>
> Lastly, we have included societal impacts in our supplementary material. We plan to move it in the main paper to emphasize on it. We will also add more discussions on the potential limitations in the final version as per your suggestion.

---

> > ### Comment · Reviewer_mRRK · 2021-09-02
> > **After rebuttal**
> >
> > Thank the authors for clarifying my concerns. The authors addressed my technical questions in the rebuttal, so I would raise my score from borderline reject to borderline accept.

---

### Official Review · Reviewer_Wohx · 2021-07-19

**Rating:** 6
**Confidence:** 4

**Summary:**

This paper proposed to use the vision transformer as the visual backbone for vision-language pre-training. To quantize the information flow transferred between vision and language modalities, the authors proposed a metric called Inter-modality flow that aggregates the attention weight across different layers. Experiments on VQA, visual entailment, and visual reasoning validates that the proposed method achieves state-of-the-art performance.

**Limitations And Societal Impact:**

The paper didn't provide texts about limitations and societal Impact.

**Main Review:**

[Strength]

- Using visual transformers as the backbone to extract visual features is a neat and intuitive idea. The whole architecture is more consistent with the multi-modal transformers. The proposed IMF seems to have correlations with the downstream task performance, which might be a good indicator to measure the pre-trained vision language model without downstream tasks.

[Weakness]

- My major concern of this paper is the lack of novelty. Visual parsing model is basically the vision transformer backbone, Masked feature regression is very similar to other pre-training objectives and has been used in previous work. The inter-modality flow can only measure the interaction between different modalities,

- For masked feature regression, the proposed mask the token with top-k attention weight, is it possible to mask based on the proposed IMF metric?

- The paper only tests with 3 downstream tasks (VQA, NLVR, Visual Entailment). Some other tasks which might require more vision-language interaction (VCR) are not tested in the paper. What is the IMF across different tasks? For example, is the IMF required in VQA different from NLVR? It will be good to add these ablation studies.

**Time Spent Reviewing:**

2 hours

---

> ### Author Response · Authors · 2021-08-10
> **Response to Reviewer Wohx**
>
> 1. **Novelty.**
> To the best of our knowledge, this is the first work studying the effect of different visual model structures (e.g., bottom-up features, CNN features, image patches, visual tokens) on multi-modal learning.
> Under this motivation, we propose (a) visual parsing and (b) MFR to achieve state-of-the-art results with even smaller image size. We propose (c) IMF as a quantifiable measurement to measure the influence of different structures on multi-modal interaction.
> We believe our work will inspire future discussion and analysis on structure design for vision-language pre-training.
>
>     (a) Visual parsing. This work first explore the vision Transformer for vision-language pre-training. The whole architecture is more consistent with the multi-modal transformers, which simplify the training procedure by unifying optimizers.
>
>     (b) Masked Feature Regression (MFR). Our MFR is the first ranking-based masking method leveraging the intrinsic properties of the visual extractor, instead of random masking. The pre-training objective of MFR (Masked Feature Regression) is effective in previous region-based pre-training works (e.g., ViLBERT, UNITER). However, MFR is still under exploration in end-to-end grid-based pre-training works. For example, the objective of MPP (Masked Patch Prediction) proposed in ViLT harms downstream tasks' performance [22]. Our ablation experiments also show that masking visual features randomly is useless. To solve this, we propose ranking-based MFR to explore using the intrinsic properties of the visual extractor to mask. Specifically, we leverage the attention scores in vision Transformer. This effective masking method makes the cross-modal Transformer attend more into the language side, leading to better inter-modal learning.
>
>     (c) Inter-Modality Flow (IMF). The IMF is a novel metric to reflect the cross-modal interactions of different vision-language frameworks. Our IMF shows high correlation with downstream performances, which validates the importance of inter-modal interaction during pre-training. Our IMF is specially designed for interaction measurement, which is under explored yet.
>
> 2. **Usage of MFR.**
> We cannot mask tokens based on our proposed IMF metric. The masking operation is conducted on the input of the cross-modal Transformer, while the IMF metric is calculated based on the output of the cross-modal Transformer to study the interaction of vision and language modalities.
>
>
> 3. **About downstream tasks.** Following existing works (e.g., ViLT [22], UNITER [9], SOHO [16]), we choose four common downstream tasks and put image-text retrieval results in supplementary materials.
> Thanks for your suggestion. We will consider evaluating IMF metric on downstream tasks to measure the fusion of inter-modality as our future work. While in this work, we aim to explore and compare how vision and language modalities interchange their information in different pre-training models (Figure 3 and Figure 4). We have checked the correlation of the IMF scores in pre-training models with the downstream performances (L295-302). The positive correlation reflects the effectiveness of IMF in measuring inter-modality in VLP.
>
> Lastly, we include societal impacts in our supplementary material, which is allowed by policies. We also include image-text retrieval results in the supplementary material due to limited space. We will move them to the main paper to emphasize more on them in final version.

---

### Decision · Program_Chairs · 2021-09-27

**Decision:**

Accept (Poster)

**Comment:**

The paper initially received divergent recommendations, but after the detailed author response all reviewers lean to accept the paper.

I recommend accept under the expectation that authors will revise the paper according to the detailed author response, addressing the reviewers' concerns. This includes, but is not limited to the following points
1) clarifications & delineations, including to prior work.
2) Additionally results and ablations (tables) provided in author response
3) details on model size in FLOPs
[some can go to supplement if they don't fit in the page limit]